# Fault Classification for Cooling System of Hydraulic Machinery Using AI

**DOI:** 10.3390/s23167152

**Published:** 2023-08-13

**Authors:** Haseeb Ahmed Khan, Uzair Bhatti, Khurram Kamal, Mohammed Alkahtani, Mustufa Haider Abidi, Senthan Mathavan

**Affiliations:** 1Department of Engineering Sciences, National University of Sciences and Technology, Islamabad 44000, Pakistan; haseebahmedkhan.92@gmail.com (H.A.K.); khurram.kamal@pnec.nust.edu.pk (K.K.); 2Department of Industrial Engineering, College of Engineering, King Saud University, P.O. Box 800, Riyadh 11421, Saudi Arabia; moalkahtani@ksu.edu.sa (M.A.); mabidi@ksu.edu.sa (M.H.A.); 3Department of Civil and Structural Engineering, Nottingham Trent University, Burton Street, Nottingham NG1 4BU, UK; b00021159@gmail.com

**Keywords:** hydraulic systems, breakdown, Artificial Intelligence, sustainable, fault conditions, spectrograms, hydraulic test rig, sensors, machine learning, deep learning

## Abstract

Hydraulic systems are used in all kinds of industries. Mills, manufacturing, robotics, and Ports require the use of Hydraulic Equipment. Many industries prefer to use hydraulic systems due to their numerous advantages over electrical and mechanical systems. Hence, the growth in demand for hydraulic systems has been increasing over time. Due to its vast variety of applications, the faults in hydraulic systems can cause a breakdown. Using Artificial-Intelligence (AI)-based approaches, faults can be classified and predicted to avoid downtime and ensure sustainable operations. This research work proposes a novel approach for the classification of the cooling behavior of a hydraulic test rig. Three fault conditions for the cooling system of the hydraulic test rig were used. The spectrograms were generated using the time series data for three fault conditions. The CNN variant, the Residual Network, was used for the classification of the fault conditions. Various features were extracted from the data including the F-score, precision, accuracy, and recall using a Confusion Matrix. The data contained 43,680 attributes and 2205 instances. After testing, validating, and training, the model accuracy of the ResNet-18 architecture was found to be close to 95%.

## 1. Introduction

According to a leading market company for global coverage, technavio, the global Hydraulic Equipment Market will accelerate its growth at a Compound Annual Growth Rate (CAGR) of 4.71% for the period of 2020–2025. It had 3.23% growth for the year 2021. The growth contributed by the Asia-Pacific (APAC) Region is 47%. The incremental growth during the period 2020–2025 is estimated to be around USD 15.50 billion [1]. Hydraulic systems are used in almost every industrial sector, and their application is widely known in major industries including manufacturing, construction, and robotics, to name a few. They are notable in industries for a reason: cost-effectiveness, efficacy, and adaptability. Many industries prefer to use hydraulic systems due to their numerous advantages over mechanical, electrical, and pneumatic systems. Their capability of moving heavier loads with sustained force and torque and providing significantly more power than electrical, mechanical, and pneumatic systems are not surprising to anyone. The advantage of a fluid power system is that it can easily get through a range of heavy loads without having to use gears, levers, and pulleys.

The increasing demand for Hydraulic Equipment requires flawless operation, and finding the source of the problem becomes a challenging task when the hydraulic system fails. This may include the motor, pump, valves, actuators, and hydraulic fluid. Human error and faulty maintenance practices can be additional sources of failure. Some of the common reasons for hydraulic failure include water and air pollution, temperature problems, the levels of fluid, and quality and human errors. These failures give rise to equipment breakdown and may cause some serious damage. According to an estimate, the breakdown of machines can be up to 20% [2].

Numerous studies have already been conducted by researchers on the fault classification of various systems including hydraulic systems. Adamsa et al. broke down the conventional problem into several subproblems under a hierarchical classification scheme to achieve maximum accuracy for Prognostic Health Monitoring (PHM). Furthermore, for each subproblem, reinforcement learning was suggested for the classifiers. They tested three reinforcement learning algorithms, i.e., Monte Carlo learning, Q-learning, and SARSAA. The hydraulic actuator’s condition was evaluated using the suggested methods [3].

Helwig et al. evaluated the classification of component conditioning using LDA by simulating fault situations such as valve changing deterioration, pump leaks, accumulator gas leaks, and oil dispersion, with the help of an experimental hydraulic test bench [4].

To tackle the issue of defect diagnosis utilizing bearing data given by the Case Western Reserve University (CWRU) data center, Yuan Y. et al. categorized the fault types using a CNN [5].

Nikolai et al. used supervised classification based on LDA for the evaluation of the statistical data acquired from the condition monitoring system. Automated feature extraction was used, and fault scenarios were diagnosed by using correlation criteria [6].

Cloud and edge servers were used by Fawwaz et al. to propose real-time fault recognition. Feature selection and offline learning were performed on the cloud, while online recognition close to the data source was computed at the edge. GA-based feature selection was used, and LSTM-AE was used as the fault detection model [7].

Keke Huang et al. discussed the problems faced in the acquisition of data in hydraulic systems such as multi-rate sensor data. To overcome the problem, a deep learning approach was proposed in their research work, which was able to automatically extract features from the data samples having multiple rates of acquisition. A CNN-based algorithm was designed to perform the classification of faults in hydraulic systems, which had a 10% better classification accuracy compared with the multiclass SVM [8].

Mallak et al. and Fathi et al. worked on the diagnostics and classification of hydraulic system faults. The proposed architecture was a combination of deep learning models, supervised machine learning models, and LSTM autoencoders. The system was tested on component faults, as well as sensor faults in a hydraulic system [9].

Silverstein et al. published research work on the comparison of different deep learning and traditional machine algorithms in the prediction of faults in hydraulic systems. They used the deep learning algorithms LSTM and TCN from, as well as machine learning algorithms: Decision Trees, k-Nearest Neighbors, and Random Forest. It was observed that TCN performed best, and enhanced feature engineering in traditional machine learning models also gave satisfactory results [10].

Yoo et al. presented an algorithm in their paper for the detection of faults using correlation-based clustering. The proposed approach was different from traditional clustering algorithms, as they decreased the size of the data to speed up the process of training the algorithm. The proposed approach clustered datasets of high correlation in a straight line; the distance of each dataset was calculated using the stochastic distance, and the abnormality detection index was also calculated. The proposed algorithm was applied to a dataset of a hydraulic system for verification [11].

Zhuo et al. and Z. Ge et al. published their research work in the field of industrial processes. The goal of the work was to develop a list of potential faults, with the power of existing data, that could occur in the future or have not been addressed before. The Generative Adversarial Network (GAN) was utilized to deal with this any-shot learning problem. The algorithm was tested on data acquired from a hydraulic system and TEP [12].

Konig et al. and Helmi et al. published a research paper in which they proposed a deep-learning-based model to provide data about the sensitivity and significance of the sensors in condition monitoring. Another objective of the research was to observe the contribution of each sensor to the overall result. Matthew’s Correlation Coefficient (MCC) was used to determine the classifier’s accuracy [13].

The research paper published by Kim et al. and Jeong et al. proposed a novel approach to fault classification in hydraulic systems. A setup was developed to track the condition of the overall system in real-time. Features were extracted from the input data using a CNN and a BiLSTM trained on the features; a sigmoid function was used as the classifier, and the data learned by BiLSTM were fed to the classifier. To overcome the problem of data scarcity, an augmentation technique was used to create new data. It was reported that the projected system performed better than the conventional deep learning approaches [14].

Yantao Zhu proposed a hybrid approach for dam deformation prediction by combining statistical regression, phase space reconstruction, and an improved LSTM neural network with parameter optimization using the GWO algorithm. The experimental results on a high-arch dam demonstrated effective noise elimination and high prediction accuracy, offering a balance between model performance and interpretability [15].

Leong et al., Ooi et al., and Lim et al. proposed an adaptive Genetic Algorithm to overcome problems faced in standard feature selection processes such as convergence towards local optima, manual parameter tuning, premature convergence, lower feature subset reduction rates, and the excessive cost of computation. The proposed STLA-GA was able to outperform classic feature-selection methods due to its adaptive nature [16].

A study conducted Bo Jin applied deep transfer learning to detect the occurrence of diseases using facial diagnosis. The top 1 accuracy was greater than 90%, which was a better accuracy than the traditional machine learning method and clinical diagnosis [17].

Qinghe Zheng conducted a study for automatic modulation classification by implementing spectrum interference based on a two-level data augmentation method. After comparison with a variety of data augmentation techniques, it was concluded that the proposed method showed much advancement [18].

Nantian Huang proposed using a label-noise-robust Auxiliary Classifier Generative Adversarial Network (rAC-GAN) for fault diagnosis of rolling bearings for wind turbine gearboxes. The model demonstrated higher accuracy than other models available for multistate classification of rolling bearings [19].

Bin Cao studied how deploying sensor nodes and relay nodes in an industrial environment affected security, lifetime, and coverage issues. The author applied six serial algorithms and two parallel algorithms. This configuration allowed the author to achieve better performance in less time [20].

Hanxin Chen conducted a study in which the author proposed a diagnosis method to detect faults in a centrifugal pump using a three-dimensional matrix having time, frequency, and space as its dimensions. This study assisted in understanding the normal and faulty states of the mechanical equipment using improved particle swarm algorithms [21].

The study conducted by Yantao Zhu proposed an automatic damage-detection method for large-volume hydraulic concrete structures using drones and AI techniques. The approach combined computer vision with the Xception backbone network for crack feature extraction and an adaptive attention mechanism based on Deeplab V3+ for precise identification of cracks, achieving high-precision results with a 90.537% Intersection Over Union (IOU), 91.227% precision, 91.301% recall, and 91.264% F1-score [22].

Data augmentation was applied to the time series data of a hydraulic system to obtain a hefty amount of data for better classification because deep learning models do not work well when the amount of data is tiny. A five-layer Convolutional-Neural-Network-based Visual Geometry Group Network (VGGNet) was used for data augmentation [23].

Kortmann et al. presented a method for feature extraction; the approach utilized an unsupervised autoencoder for the task. The state of the cooler was set as the target variable from the dataset for classification purposes and regression, and the hydraulic accumulator pressure was set. It was reported that the prediction accuracy was not as good as other approaches currently available for performing the same tasks [24].

To determine the condition of a hydraulic system, Y. Cheng et al. suggested a technique based on General multi-class Support Vector Machines (GenSVMs). The statistical features from the raw data were extracted during preprocessing such as the skewness, kurtosis, absolute mean value, and root mean square value. Four different GenSVMs were developed for four dissimilar components of the hydraulic system: pump, accumulator, cooler, and valve. The proposed model achieved a 100% classification accuracy for the pump, cooler, and valve and 76% for the accumulator. The proposed GenSVMs showed a better classification accuracy than the four other ML algorithms it was compared with [25].

ResNet-18 is a revolutionary deep convolutional neural network architecture introduced by Microsoft Research. It is differentiated from other methods through the incorporation of residual connections, also known as skip connections, which enable the learning of residual mappings and simplify the training of deeper networks. With 18 layers, including convolutional, pooling, and fully connected layers, ResNet-18 overcomes the vanishing gradient problem, making it easier for the network to learn effectively. The use of shortcut connections ensures that the input and output dimensions of each residual block remain the same, and additional 1 × 1 convolutional layers are introduced when necessary to maintain consistency. Despite its depth, ResNet-18 is computationally efficient with fewer parameters. Pre-trained models on large datasets, such as ImageNet, are often utilized for transfer learning, making ResNet-18 a widely adopted and powerful model for various image-recognition tasks, with a significant impact on subsequent deep learning architectures. ResNet-18 is relatively deeper compared to earlier CNN architectures such as AlexNet or VGG-16. It has 18 layers, including convolutional layers, pooling layers, and fully connected layers, while traditional CNNs typically have fewer layers.

In this paper, the authors propose the prediction of the working behavior of cooling circuits in a hydraulic system. XGBoost and ReliefF were compared for the feature ranking technique. XGBoost is a library that implements machine learning algorithms under the Gradient Boosting framework, while ReliefF is an algorithm that takes a filter-method approach to feature selection to calculate feature scores to rank the top-scoring features. A DNN and ANN were used for the prediction of the cooling circuit [26].

The study aims to investigate techniques used for the categorization of Hydraulic Systems; furthermore, it explains a new technique for the classification of the fault conditions of hydraulic systems using an Artificial-Intelligence-based approach.

The dataset was prepared using the test rig. The rig comprised multiple sensors including pressure sensors, volume flow sensors, temperature sensors, and various sensors for monitoring motor power, cooling efficiency, vibration, cooling power, and system efficiency. It included failure scenarios that depict the fault state of four primary components. The data were analyzed using deep learning technology, comprising 43,680 attributes and 2205 instances. Thirty percent of the data were tested and validated, after which the data were trained on for classification. A hydraulic test rig was used to experimentally gather the dataset for this study. A primary working circuit and a secondary cooling–filtration circuit made up the test rig, and they were connected by an oil tank. The system measured process parameters such as pressures, volume flows, and temperatures, while cycling through constant load cycles that lasted 60 s. Throughout the testing, the state of four hydraulic components, i.e., the cooler, valve, pump, and accumulator, was quantitatively changed.

The primary working circuit was in charge of sending energy from the actuator to the pump. It was made up of an actuator, a valve, and a pump. The hydraulic fluid was cooled and filtered via the secondary cooling–filtration circuit. It was made up of a reservoir, a filter, and a cooler. The hydraulic fluid was kept in the oil tank. The fluid was also cooled and filtered as a result. Constant load cycles simulated the types of loading that a hydraulic system would encounter in a practical application. The process values measured included pressures, volume flows, and temperatures. These numbers were used to evaluate the hydraulic system’s efficiency and the health of its parts.

The dataset contains 43,680 attributes and 2205 instances. The attributes are the features of the dataset, and they represent the values that were measured by the sensors on the hydraulic test rig. The instances represent a single constant load cycle of 60 s in various conditions.

The attributes in the dataset include the pressures, volume flows, temperatures, and the condition of four hydraulic components: the cooler, valve, pump, and accumulator. The faults in the dataset represent intentional changes that were made to the hydraulic system to simulate the different types of faults that could occur in a real-world system. These faults contributed to the process sensor measurements, which could be used to assess the performance of the hydraulic system and the condition of the hydraulic components.

The dataset is a valuable resource for researchers who are interested in studying the performance of hydraulic systems. The dataset can be used to train machine learning models that can be used to predict the performance of hydraulic systems under different operating conditions. The dataset can also be used to develop new fault detection and diagnosis methods for hydraulic systems. Other approaches have been applied by researchers to evaluate prediction accuracy for their study. In this study, ResNet-18 was used to evaluate the prediction accuracy. Hence, this study aimed to achieve higher prediction accuracy than other techniques available for deep learning. In Table 1, the predication accuracies for different models are analyzed. This was used as a reference for the comparison of the prediction accuracy of ResNet-18.

## 2. Materials and Methods

The hydraulic system dataset is open source and can be obtained from the University of California Irvine (UCI) machine learning repository [28]. The data were prepared using the test rig. The rig was comprised of multiple sensors including pressure sensors, volume flow sensors, temperature sensors, and various sensors for monitoring motor power, cooling efficiency, vibration, cooling power, and system efficiency. Constant load cycles of 60 s were repeated cyclically for measuring the process values, i.e., pressures, temperatures, and volume flows. Additionally, the dataset includes failure scenarios that depict the fault state of four primary components: cooler, valve, internal pump leakage, and accumulator.

The information provided by the UCL repository is in several text files. As shown in Figure 1, six distinct text files include the processed sensor readings and one file contains the intended fault circumstances.

The data were combined into a single data frame after being translated to comma-separated values. Using the filter option in Microsoft Excel, different files for the fault conditions of the cooler system of the hydraulic test rig were developed by extracting the data from the .csv file, as demonstrated in Figure 2.

A spectrogram is a visual representation of the signal strength over time at different frequencies. A spectrogram is created using the Fast Fourier Transform by passing through a digital process. The digitally sampled data are divided into blocks in the time domain, which are overlapping, and Fourier Transformed to determine the size of the frequency spectrum for each block. Then, for each block, a vertical line in the image is represented; it is the midpoint of the block, which corresponds to the magnitude versus frequency for a specific moment. The time plots are then arranged side by side to create an image or three-dimensional surface and can be windowed in a variety of ways with a small amount of overlap. The computation of the signal’s Short-Time Fourier Transform and this operation are related [29].

A continuous signal can be subjected to sampling to transform it into a discrete time signal by selecting values from the continuous time signal at evenly spaced points in time [27]. Hence, sampling a continuous-time signal *x* with sampling period *T_s_* provides the discrete time signal *x_s_*, defined by:(1)$$x_{s}(n)=x(nT_{s})$$

Angular frequency sampling is given by:(2)$$\omega_{s}=2\pi /T_{s}$$

A total of 440 spectrograms were generated for the targeted fault conditions of the cooler conditions: 146 spectrograms for Class Output 3 depicting the fault condition “close to failure” behavior of the cooler condition, 146 spectrograms for Class Output 20 depicting the fault condition “reduced efficiency” behavior of the cooler condition, and 148 spectrograms for Class Output 100 depicting the condition “full efficiency” behavior of the cooler condition.

AI by definition is the effort to automate intellectual tasks normally performed by humans (DL-w-P). It is a field of study that has enabled mankind to ponder how to integrate and utilize information, process and analyze data, and leverage the power of data to enhance the decision-making abilities of machines. The power of AI is being utilized in a variety of sectors, such as finance, criminal justice, supply chains, automobiles, search engines, robotics, etc. [30].

ML is a major field of AI, and according to Tom Mitchell, “Machine learning is the study of computer algorithms that allow computer programs to automatically improve through experience” [31].

ML algorithms are majorly applied for regression, classification, clustering, recommender systems, and dimensionality reduction.

DL is regarded as an advancement of ML in which programmable NNs empower machines to make a judgment in the absence of human involvement. DL models are created such that they continually examine the data using a logical framework and behave similarly to humans’ decision-making abilities. In this regard, DL applications work on a complex structure of algorithms called Artificial Neural Networks (ANNs). The biological network of neurons in the human brain has influenced the design of ANNs. The learning systems of ANNs are considerably more competent than those of classical ML models. The DL model cannot make erroneous conclusions. However, similar to most AI cases, it also needs several trainings to ensure the learning process is true. However, when executed completely and properly, deep learning is shown to be a scientific marvel, and it is considered the backbone of true AI [32]. There are several DL algorithms used in DL models. Some of the widely used algorithms are CNNs, LSTM, RNNs, MLPs, SOMs, and autoencoders.

Residual Network-18 (ResNet-18) is a Convolutional Neural Network that is 18 layers deep.

The convolutional layer carries out the largest portion of the network’s computational burden. This layer receives the picture as the input and applies various convolutional procedures to it. It has several filters, sometimes known as kernels, and all of the training covers the parameters. In general, the filters are smaller than the original image. To create an activation map, the filters are convolved with each picture. Every component of the filter and image dot product is computed at every place as the filter advances over the picture’s width and height. This process is repeated for every element of the input layer to generate an activation map [33]. The main objective of the convolutional layer is the extraction of information and patterns from an image. Filters or kernels are in charge of retrieving low-level characteristics such as the color, gradient direction, etc., from the beginning of the networks. On the other hand, higher-level characteristics such as image edges are extracted by filters or kernels farther down the network.

The weighted sum of the input from a node or nodes is transformed into the output in a layer of the NN by such a function. The performance and capability of the neural network mainly depends on the choice of such a function. Different functions are needed for various model components. In most cases, networks are built so that each node in a layer uses the same function. However, such a function is used either following or during the internal processing of each network node.

A network normally consists of layers: an input layer in which raw data are fed from the domain, hidden layers that pass the output to an additional layer by taking the input from another layer, and output layers responsible for the predictions. Typically, the same activation function is used by all hidden layers. Different activation functions from the hidden layers are used by the output layer, and this depends on the type of prediction required by the model.

CNNs are specially designed for object detection or image processing. CNNs are normally used to empower computer vision by teaching the machine about processing the visual world. Facial Recognition Technology is one of the common uses of computer vision. To make it simple, a binary representation of the data is given as the digital picture. It has a collection of pixels that are organized in a grid-like pattern and carry data about each pixel, including its color and brightness.

The pooling layer receives the corrected feature map as the input. A series of these operations, max pooling to be precise, on an image is performed by the pooling layer. The dimensions of the feature map are reduced by the operation of down-sampling via pooling. This reduces the amount of computation and weights needed in the network. The resultant two-dimensional array from the pooled feature map is flattened into a single, continuous, linear vector by the pooling layer [34]. To make the network more flexible, pooling merges many pixel values into a single one. This helps to decrease the chances of over-fitting, biasing networks towards particular pixels [35].

The Confusion Matrix technique is used to describe the summary of the performance measurement of a machine learning categorization. It gives a better idea about the classification model regarding whether making the right predictions and the measure of the errors it made. The performance of two or more classes can be measured by the amount of correct and faulty forecasts, a count value summary that is split into the classes.

It shows the confusion of the classification models while making predictions. It not only presents insights into the errors, but also the types of errors made by the classifier:

TP = number of True Positives;

FP = number of False Positives;

TN = number of True Negatives;

FN = number of False Negatives.

Following are the performance measures that can be observed by the Confusion Matrix.

The percentage of accurate forecasts with respect to all other forecasts is known as the accuracy. It is calculated using the formula:(3)$$Accuracy\left(\%\right)=\frac{\left(TP+TN\right)}{\left(TP+TN+FN+FP\right)}\times 100$$

It evaluates the binary classification system by classifying the example into “positive” and “negative”. The F-score is a measure that combines the model’s recall and precision. It can be regarded as the “Harmonic Mean” of the model. It can be calculated using the formula:(4)$$F-Score\left(\%\right)=\left(\frac{2\times \left(Recall\times Precision\right)}{\left(Recall+Precision\right)}\right)\times 100$$

The recall is the proportion of correctly predicted positive outcomes to all positive outcomes. It can be calculated using the formula:(5)$$Recall\left(\%\right)=\frac{TP}{\left(TP+FN\right)}\times 100$$

The precision is the ratio of accurate positive forecasts to all positive predictions. It can be calculated using the formula:(6)$$Precision\left(\%\right)=\frac{TP}{\left(TP+FP\right)}\times 100$$

Residual Networks (ResNets) are extremely successful due to their relative improvement of around 28%, replacing the other architectures. They can be efficiently trained with 100 layers, as well as 1000 layers [36]. Stacking additional layers in Deep Neural Networks works efficiently with improved accuracy and performance for solving complex problems. The ResNet architecture was adopted for the fault classification of hydraulic systems.

The model was trained using MATLAB with 20 epochs for each iteration with the pool size being 3,3. Table 2 describes the parameters from the hydraulic test rig that were analyzed. Table 3 classifies the fault conditions of the hydraulic test rig depending on the class output.

The performance of a machine learning model is assessed using evaluation metrics. They are commonly employed for assessing a model’s performance over time or to compare different models.

The following are some of the most-popular evaluation metrics:

Accuracy: This is the proportion of predictions that are valid. Although it is the easiest metric to comprehend, it occasionally contains errors. For instance, even if a model is not particularly effective at forecasting the minority class, it will have a high accuracy if it consistently forecasts the majority class.

Precision: The percentage of accurate positive predictions is represented by this number. A model might predict that 100 out of 1000 cases will be positive, and if 90 of those predictions come true, then its precision is 90%.

Recall: This represents the percentage of real positives that were accurately anticipated. For instance, if the dataset contains 100 actual positives and the model accurately predicts 90 of them, the recall is 90%.

F-score: This represents the weighted average of the recall and precision. It is frequently used as a lone metric to sum up a model’s performance.

The Confusion Matrix is a table that displays the quantity of true positives, false positives, false negatives, and true negatives in order to summarize the performance of a model. It can be used to determine the F-score, accuracy, precision, and recall.

It is impossible to exaggerate the significance of evaluation metrics. They are crucial for comprehending a machine learning model’s performance and for making defensible choices regarding how to enhance the model.

Additional information on the significance of evaluation measures is provided below:

For various tasks, different metrics are appropriate. For activities where the cost of false positives and false negatives is comparable, accuracy, for instance, is an appropriate statistic. However, accuracy is a better indicator for activities when the cost of false positives is significantly larger than the cost of false negatives.

It is crucial to take into account not just the accuracy, but also the full Confusion Matrix. The Confusion Matrix can shed light on the kinds of mistakes the model is making. For instance, if a model has high accuracy, but low recall, it is probably underestimating the number of positive examples.

It is crucial to analyze a model using a variety of metrics. There is no one metric that can fully capture the performance of the model. One can have a better grasp of the model’s advantages and disadvantages by using a variety of measurements.

## 3. Results

After all the spectrograms were generated, they were classified into three categories: close to failure, reduced efficiency, and full efficiency. Close to failure was assigned a class output of 3; similarly, reduced efficiency and full efficiency were assigned class outputs of 20 and 100, respectively. Among the spectrograms, 146 were Class Output 3, 148 were Class Output 20, and 146 were Class Output 100 out of a total of 440 spectrograms. Figure 3, Figure 4 and Figure 5 show the sample spectrograms with operational, close to failure, and failed hydraulic cooling system.

The color of each point on the spectrogram represents the power of the radiation at that frequency and time. The spectrogram shows the power of the radiation from the hydraulic test rig. The different frequencies in the radiation can be attributed to the different components of the hydraulic system. The variation in the power of the radiation over time can be attributed to the different fault conditions of the cooling circuit of the hydraulic system. Each figure has a slight difference from the others. In Figure 4, a sharp green line can be observed in the third power radiation; however, all other power radiations are identical. In Figure 5, the 3rd and 4th power radiations include green lines that are not as sharp as those in Figure 4; however, the rest of the power radiations are identical, which makes it different from the other figures.

The Confusion Matrices shown in Figure 6, Figure 7 and Figure 8 were generated for the training, validation, and testing of the model of the cooling system for the hydraulic test rig, respectively.

In the training case, the overall accuracy of the model was found to be 98%. The model could correctly distinguish 98% of the fault condition “reduced efficiency” and 99% of both the “close to total failure” and “full efficiency” conditions of the cooling system of the hydraulic system. The results showed the high accuracy of the training of the model.

The overall precision of the training model was calculated as 98%. The precision for the fault condition “close to total failure” was 100%. The precision for the “reduced efficiency” fault condition was 97% and for the “full efficiency” condition of the cooling system of the hydraulic system was 98%.

The overall recall of the training model was calculated as 98%. The recall for the condition “full efficiency’’ was 100%. The recall for the “reduced efficiency” fault condition was 98% and for the “close to total failure” fault condition of the cooling system of the hydraulic system was 97%.

The overall F-score of the training model was calculated as 98%. The F-score for the condition “full efficiency’’ was 99%. The F-score for the “reduced efficiency” fault condition was 97% and for the “close to total failure” fault condition of the cooling system of the hydraulic system was 98%.

For the validation case, the accuracy, precision, recall, and F-score were determined for different conditions of the cooling system of the hydraulic test rig.

The overall accuracy obtained for the validation case was 96%. The model could predict correctly 94% of the fault conditions for the close to total failure conditions. The accuracy for reduced efficiency and full efficiency was 97% for the cooling system of the hydraulic test rig.

The overall precision of the training model was calculated as 93%. The precision was 95%, 91%, and 95% for reduced efficiency, full efficiency, and close to failure, respectively.

The precision for the fault condition “full efficiency” was 91%. However, the precision for the “reduced efficiency” fault condition was 95% and for the “close to total failure” condition of the cooling system of the hydraulic system was 95%.

The overall recall of the training model was calculated as 94%. The recall was 95%, 87%, and 87% for the reduced efficiency, full efficiency, and close to failure, respectively.

The F-score of the training model was calculated as 93%. The F-score was 91%, 95%, and 95% for the conditions close to total failure, full efficiency, and reduced efficiency, respectively.

Similarly, in the case of testing, the accuracy, precision, recall, and F-score were calculated with the help of the Confusion Matrix.

For the testing, the overall accuracy obtained was 96%. The accuracy obtained for close to failure, reduced efficiency, and full efficiency was 95%, 100%, and 95%, respectively.

The overall precision obtained was 96%. The precision obtained for close to failure, reduced efficiency, and full efficiency was 91%, 100%, and 95%, respectively.

The total percentage recall for the testing obtained was 95%. The precision obtained for close to failure, reduced efficiency, and full efficiency was 91%, 100%, and 95%, respectively.

The overall F-score of the training model was calculated as 95%. The F-score obtained for close to failure, reduced efficiency, and full efficiency was 93%, 100%, and 93%, respectively. Figure 9 shows the distribution of the TP, TN, FP, and FN for each stage. Furthermore, Figure 10 shows the Confusion Matrix according to the class output for the training, testing, and validation.

## 4. Discussion

After the literature review, different authors’ models’ accuracy were studied. Each author applied his/her technique for predicting fault conditions in the hydraulic test rig. However, in most cases, the model accuracy was not sufficient for predicting fault conditions.

The future aim of the research is to explore more suitable techniques and methods that could give greater accuracy. The impacts and results can be drawn by applying different techniques and comparisons of these results. The proposed technique can be applied for the classification of other faults of the hydraulic system, and the results can be drawn and compared. However, the classification of different sensors contributing to the condition monitoring of the hydraulic test rig can be used to conclude on the most-suitable and -reliable data of the sensors contributing to the result accuracy. ML techniques can be applied to the real-time data values of various systems for analysis. Real-time data can be processed, and more conclusions can be made. With time, more data can be extracted, and more-powerful predictions of fault conditions based on these data can be made, which can be a game changer in the Industrial Revolution.

## 5. Conclusions

The data used in the thesis work were acquired from the UCI machine learning repository. The dataset includes failure scenarios that depict the fault state of four primary components of the hydraulic test rig: cooler, valve, pump leakage, and accumulator. The fault condition of the cooler was observed for classification in this thesis work. The data are in raw text form in different files. They were preprocessed to merge them into a single file. Spectrograms were then generated for each fault condition with the help of the data provided. These spectrograms were used in the Resnet-18 architecture of Convolution Neural Networks for the classification. The results showed the high accuracy of the model, resulting in 96%, whereas the precision, recall, and F-score resulted in being 95%.

## Figures and Tables

**Figure 1 sensors-23-07152-f001:**
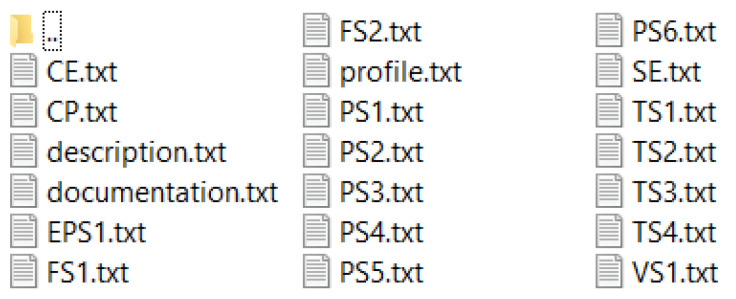
Different text files for processed sensor readings.

**Figure 2 sensors-23-07152-f002:**
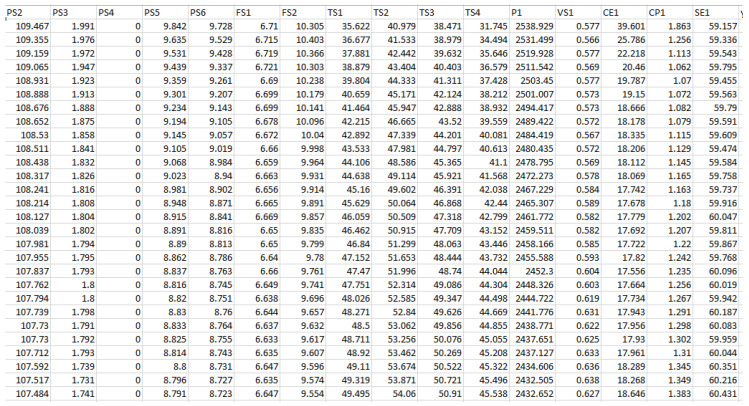
Data compiled from different sensors’ readings into a single .csv file for processing.

**Figure 3 sensors-23-07152-f003:**
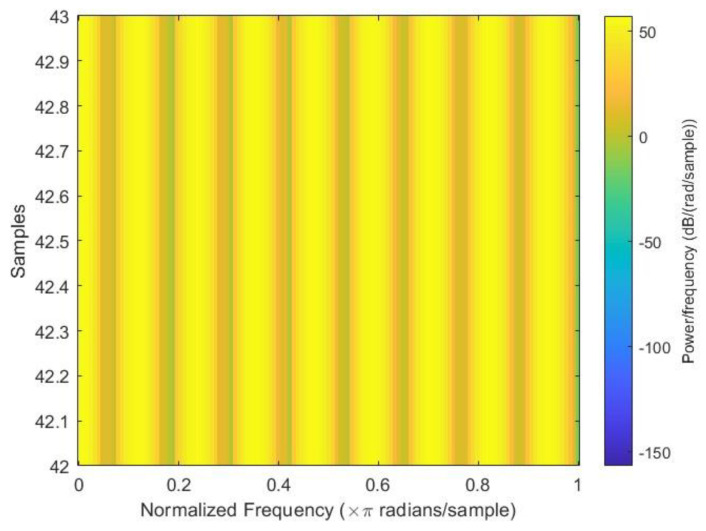
Sample spectrogram depicting the fault condition “close to failure” of the cooling circuit of the hydraulic test rig.

**Figure 4 sensors-23-07152-f004:**
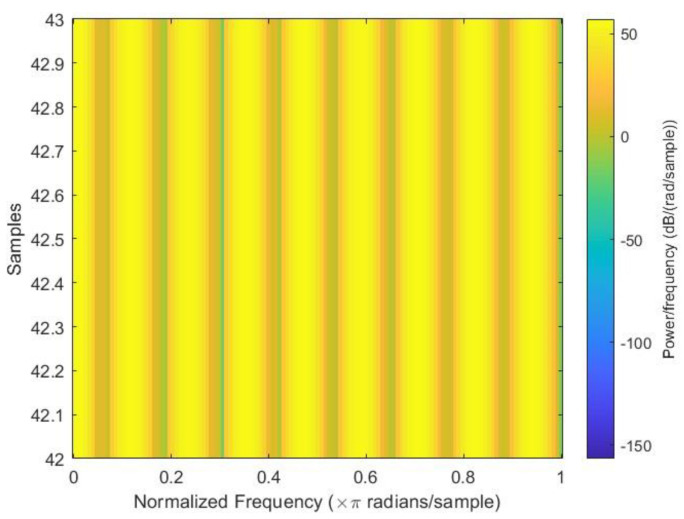
Sample spectrogram depicting the fault condition “reduced efficiency” of the cooling circuit of the hydraulic test rig.

**Figure 5 sensors-23-07152-f005:**
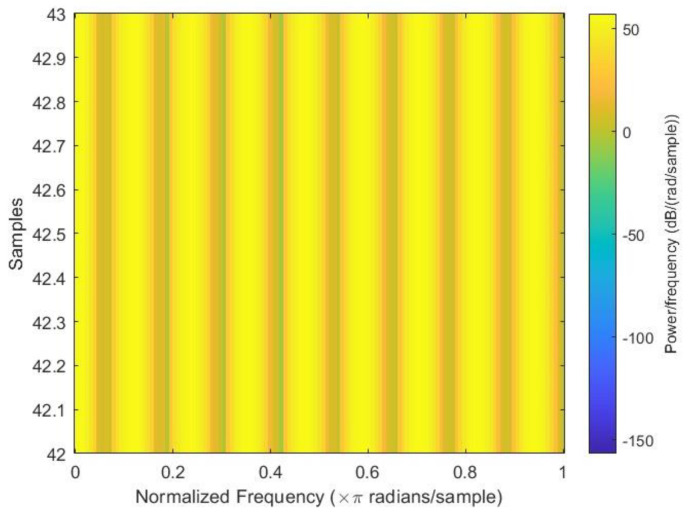
Sample spectrogram depicting the condition “full efficiency” of the cooling circuit of the hydraulic Test rig.

**Figure 6 sensors-23-07152-f006:**
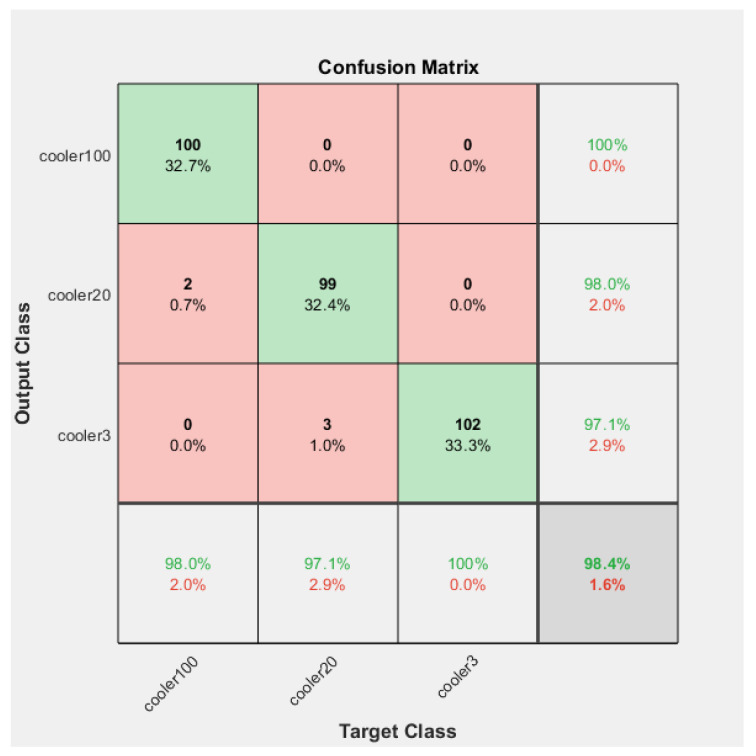
Confusion Matrix of the training of the model of the fault condition of the cooling system of the hydraulic test rig.

**Figure 7 sensors-23-07152-f007:**
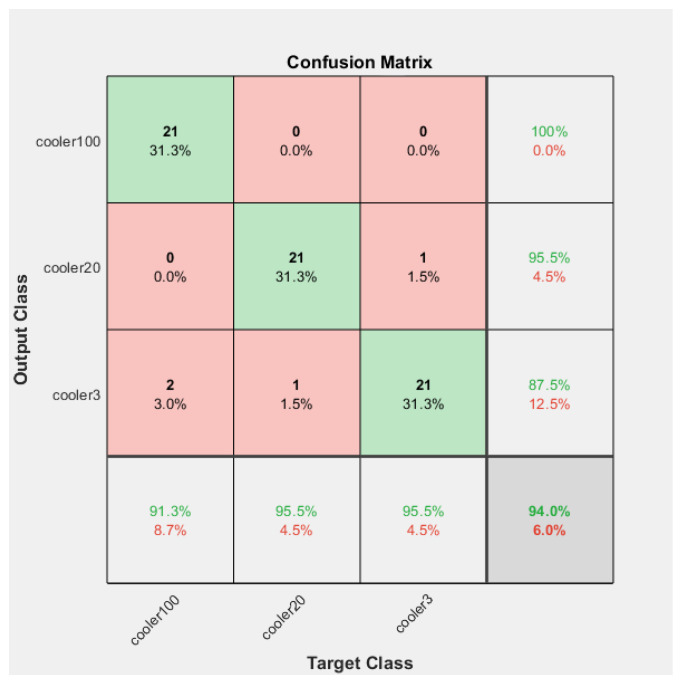
Confusion Matrix of the validation of the model of the fault condition of the cooling system of the hydraulic test rig.

**Figure 8 sensors-23-07152-f008:**
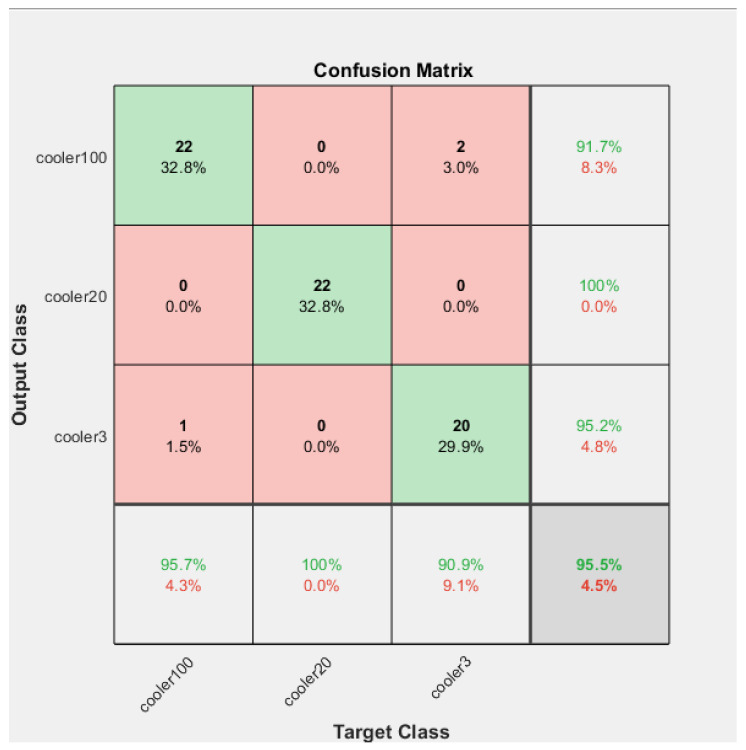
Confusion Matrix of the testing of the model for the fault condition of the cooling system of the hydraulic test rig.

**Figure 9 sensors-23-07152-f009:**
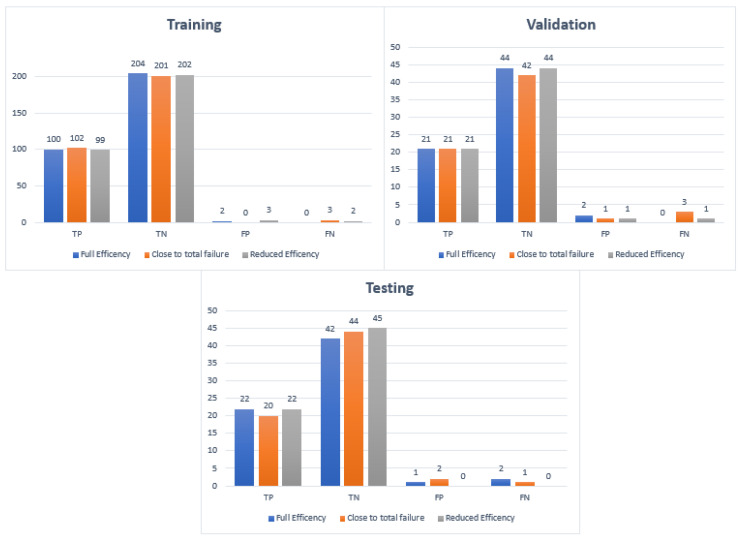
Sample distribution for TP, TN, FP, and FN for training, testing, and validation.

**Figure 10 sensors-23-07152-f010:**
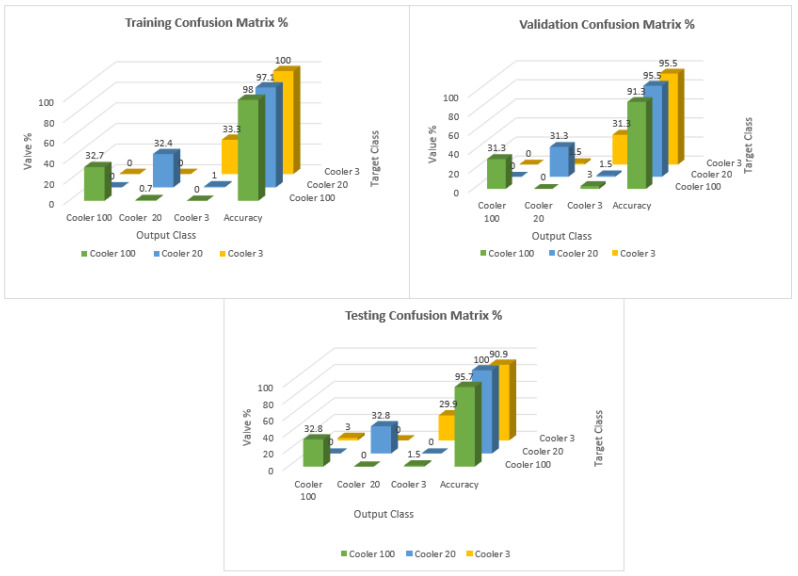
Distribution and accuracy of the samples in the Confusion Matrix depending on the class output for training, testing, and validation.

**Table 1 sensors-23-07152-t001:** Different approaches applied on the hydraulic test rig data for fault classification and its accuracy.

Approach	Application	Prediction Accuracy	Researchers
PCA and XGBoost	Fault diagnosis for hydraulic directional valves	88%	Y. Lei et al. [9]
EGMSVM	Health evaluation of complex degradation systems	94.1%	Jun Wu et al. [10]
kNN and SVM	Fault diagnosis for hydraulic systems	96.7%	X. Zhao et al. [22]
Gen-SVM	Health estimation of hydraulic systems	94%	Y. Cheng et al. [27]

**Table 2 sensors-23-07152-t002:** Parameters from the hydraulic test rig.

Pressure Sensor Data	Temperature Sensor Data	Volume Flow Data	Pump Efficiency, Cooling Efficiency, Vibration, and Efficiency Factor
PS1	TS1	FS1	EPS1
PS2	TS2	FS2	CE
PS3	TS3		CP
PS4	TS4		VS1
PS5			SE
PS6			

**Table 3 sensors-23-07152-t003:** Details of the fault condition of the hydraulic test rig depending on the class output.

Fault Type	Unit	No. of Classes	Class Output	Remark
Cooler condition	%	3	3	Close to total failure
			20	Reduced efficiency
			100	Full efficiency
Valve condition	%	4	73	Close to total failure
			80	Severe lag
			90	Small lag
			100	Optimal switch behavior
Accumulator	bar	4	90	Close to total failure
			100	Severely reduced pressure
			115	Slightly reduced pressure
			130	Optimal pressure
Internal pump leakage	-	3	0	No leakage
			1	Weak leakage
			2	Severe leakage

## Data Availability

The data presented in this study are available upon request from the corresponding author.

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
