# Peer review of "Fault Classification for Cooling System of Hydraulic Machinery Using AI"

_sensors, 2023, doi:10.3390/s23167152_

Round 1

Reviewer 1 Report

The following comments must be carefully revised.

1. The summary and description of related work in the field are insufficient. Since deep learning methods have achieved remarkable results in the field of Engineering. The following related work of deep learning must be cited and discussed, including Deep transfer learning from face recognition to facial diagnosis. IEEE Access, vol. 8, pp. 123649-123661, 2020. Spectrum interference-based two-level data augmentation method in deep learning for automatic modulation classification, Neural Computing & Applications, vol. 33, pp. 7723-7745, 2020. Fault Diagnosis of Bearing in Wind Turbine Gearbox Under Actual Operating Conditions Driven by Limited Data With Noise Labels. IEEE Transactions on Instrumentation and Measurement, 2021, 70, 1-10. Security-Aware Industrial Wireless Sensor Network Deployment Optimization. IEEE Transactions on Industrial Informatics, 2020, 16(8), 5309-5316. Multi-Sensor Data Driven with PARAFAC-IPSO-PNN for Identification of Mechanical Nonstationary Multi-Fault Mode. Machines. 2022, 10(2):155.

2. The second part should be reorganized, lacking technical details for describing methods and models, including AI model structure and training technique.

3. The theoretical basis for the proposed AI method should be supplemented. How does each module help with task performance?

4. Figure 2 and Figure 3 seem to have little difference, and the author should analyze them in more detail

5. The experiment also needs to be supplemented, lacking comparison and argumentation. Self ablation experiments are also encouraged.

6. The overall process of the methods involved is suggested to be demonstrated.

Minor editing of English language required.

Author Response

The summary and description of related work in the field are insufficient. Since deep learning methods have achieved remarkable results in the field of Engineering. The following related work of deep learning must be cited and discussed, including Deep transfer learning from face recognition to facial diagnosis. IEEE Access, vol. 8, pp. 123649-123661, 2020. Spectrum interference-based two-level data augmentation method in deep learning for automatic modulation classification, Neural Computing & Applications, vol. 33, pp. 7723-7745, 2020. Fault Diagnosis of Bearing in Wind Turbine Gearbox Under Actual Operating Conditions Driven by Limited Data With Noise Labels. IEEE Transactions on Instrumentation and Measurement, 2021, 70, 1-10. Security-Aware Industrial Wireless Sensor Network Deployment Optimization. IEEE Transactions on Industrial Informatics, 2020, 16(8), 5309-5316. Multi-Sensor Data Driven with PARAFAC-IPSO-PNN for Identification of Mechanical Nonstationary Multi-Fault Mode. Machines. 2022, 10(2):155.

Thank you for suggestion. It is important to discuss engineering work conducted in similar research domain. We have included, cited and discussed all the provided related work in detail

  1. The second part should be reorganized, lacking technical details for describing methods and models, including AI model structure and training technique.

Sure. Now we have elaborated and provided reason for the use the evaluation metric in the methods and model section. Use of their implication in practical application have been discussed now. Also comparative study in literature review have been added.

  1. The theoretical basis for the proposed AI method should be supplemented. How does each module help with task performance?

We have attempted to explain how each part of the module helps with the task performance. By including additional figures explaining the process. And further we have elaborated how the data had been acquired and how it was converted into spectrogram to generate confusion matrix for obtaining prediction accuracy and other evaluation metrics for the data.

  1. Figure 2 and Figure 3 seem to have little difference, and the author should analyze them in more detail

There are subtle changes in figure 2 and 3. There is variation in radiation of the graph in Figure 3, 4 and 35. All the details in radiation of the spectrogram have been now explained in detail. Thank you for pointing this out.

  1. The experiment also needs to be supplemented, lacking comparison and argumentation. Self-ablation experiments are also encouraged.

Thank you so much for this suggestion. We have procured vibration analyzer equipment. This would allow us to gather data related for real time data for vibration monitoring which could be used to carry out similar study. For future study we plan to carry out self-ablation test

  1. The overall process of the methods involved is suggested to be demonstrated.

Yes. Now we have demonstrated and highlighted how the data was analyzed and prepared into csv file. Along with the explanation of the process in material and method. Details for the process are also highlighted and significance is provided for overall role of it in this study.

Reviewer 2 Report

The authors’ research on fault classification of hydraulic machinery using AI presents an intriguing approach with potential practical implications. It is an interesting work and will be very helpful in related industries. There are some comments that should be clearly addressed in the manuscript.

1. The authors should carefully proofread the manuscript to improve the English expressions.

2. Figure 7 and 8 should be replotted for clear presentation.

3. There are very recent deep learning techniques that can be used for analysis tasks. The authors should mention some deep learning methods in the introduction.  doi.org/10.3390/rs15030615; doi.org/10.3390/math11092010.

4. The author mention that various evaluation metrics, including F-score, precision, accuracy, and recall, were calculated using a confusion matrix. It would be beneficial to explain the importance of these metrics in assessing the performance of the classification model and discuss their implications for practical applications. Additionally, it would be useful to mention any baselines or comparative approaches used to validate the effectiveness of your proposed approach. 

5. The dataset contains 43,680 attributes and 2,205 instances. It would be helpful to provide more context regarding the nature of these attributes and instances. For example, are these attributes derived from sensor measurements, and are the instances representative of different operating conditions or fault scenarios? 

6. It would be valuable to mention the rationale behind selecting ResNet-18 and discuss any prior work that supports its suitability for fault classification in hydraulic machinery. 

7. The author needs to supplement the comparison between the proposed model and the existing model to highlight the advantages of the proposed model, rather than simply comparing indicators.

8. It is better to simplify presentation of abstracts and introductions to highlight core content and contributions. It would be beneficial to provide more details on the specific aspects that make your approach innovative compared to existing methods. This could include highlighting any unique data collection techniques, feature extraction methods, or the utilization of AI algorithms. 

Author Response

  1. The authors should carefully proofread the manuscript to improve the English expressions.

Thank you for your suggestion. Article have been revisited and wherever English expression improvement has been done.

  1. Figure 7 and 8 should be replotted for clear presentation.

Thank you for pointing this out. We have resized the image to make its detail visible.

  1. There are very recent deep learning techniques that can be used for analysis tasks. The authors should mention some deep learning methods in the introduction. doi.org/10.3390/rs15030615;

Sure both paper have been cited since they propose recent advances in deep learning techniques. Recent paper of 2023 like these would also be considered for upcoming research projects.

  1. The author mention that various evaluation metrics, including F-score, precision, accuracy, and recall, were calculated using a confusion matrix. It would be beneficial to explain the importance of these metrics in assessing the performance of the classification model and discuss their implications for practical applications. Additionally, it would be useful to mention any baselines or comparative approaches used

Thank you for pointing this out. We have now provided detail about the significance of the evaluation metrics for our study. Their importance has been now explained in detail. 

  1. The dataset contains 43,680 attributes and 2,205 instances. It would be helpful to provide more context regarding the nature of these attributes and instances. For example, are these attributes derived from sensor measurements, and are the instances representative of different operating conditions or fault scenarios?

  1. It would be valuable to mention the rationale behind selecting ResNet-18 and discuss any prior work that supports its suitability for fault classification in hydraulic machinery.

ResNet have now been elaborated in detail in introduction section. Comparison of ResNet 18 with other architecture have also been included.

  1. The author needs to supplement the comparison between the proposed model and the existing model to highlight the advantages of the proposed model, rather than simply comparing indicators.

Thank you. Now at the end of the introduction section, the proposed model objective has been highlighted to provided to achieve higher prediction accuracy. Also, existing model accuracy have been provided for comparison with the obtained prediction accuracy for this study.

  1. It is better to simplify presentation of abstracts and introductions to highlight core content and contributions. It would be beneficial to provide more details on the specific aspects that make your approach innovative compared to existing methods. This could include highlighting any unique data collection techniques, feature extraction methods, or the utilization of AI algorithms.

    Sure. We have now included baselines and comparative section to further signify the improvement in prediction accuracy than other studies. It has been added in literature review.  ResNet have been explained a bit in abstract. It’s detail and other comparison with other architectures have been included in literature review as well.

Reviewer 3 Report

The manuscript “Fault Classification of Hydraulic Machinery using AI: A Sustainability Approach” is of interest but it is not written clearly. It is very hard to follow main point of this manuscript. What is the purpose of the part of the title “A Sustainability Approach”? Maybe is better “Fault Classification of Hydraulic Machinery using AI: Cooling behavior of a Hydraulic Test rig”. Is the manuscript about failure of cooling of the hydraulic machinery?

What is CNN variant ResNet? This term is in Abstract but the meaning is not clear. What are attributes and what are instances?

What is Technavio from Introduction?

Starting from the third paragraph “Numerous studies…”, literature overview starts which should be separate section. Btw., this literature overview is prepared in a very poor way. Hopefully, the literature overview finished with the paragraph staring with “In this paper, author proposed…” Anyway, it discussed two unknown terms: XGBoost and ReliefF.

Main part is in “Materials and Methods” which contains possible contribution, but anyway it is unclear.

References are not in correct format.

Author Response

The manuscript “Fault Classification of Hydraulic Machinery using AI: A Sustainability Approach” is of interest but it is not written clearly. It is very hard to follow main point of this manuscript. What is the purpose of the part of the title “A Sustainability Approach”? Maybe is better “Fault Classification of Hydraulic Machinery using AI: Cooling behavior of a Hydraulic Test rig”. Is the manuscript about failure of cooling of the hydraulic machinery?

Thank you for your suggestion. We have updated the name of the research work to          Fault Classification for Cooling System Of Hydraulic Machinery using AI Purpose of using sustainability was to highlight the role of this technology for condition based maintenance.

What is CNN variant ResNet? This term is in Abstract but the meaning is not clear. What are attributes and what are instances?

ResNet is abbreviation for Residual Network in Abstract ResNet has been now elaborated. It can train upto 1000 layers. Detail explanation for ResNet have also been added to the literature review as well.

What is Technavio from Introduction?

Thank you for pointing this out. It is actually a leading market research company. Details are specified in

Starting from the third paragraph “Numerous studies…”, literature overview starts which should be separate section. Btw., this literature overview is prepared in a very poor way. Hopefully, the literature overview finished with the paragraph staring with “In this paper, author proposed…” Anyway, it discussed two unknown terms: XGBoost and ReliefF.

Sorry for missing this point. XGBoost is a machine learning library used for regression, classification and ranking problem. ReliefF is an algorithm which is used similarly like XGBoost for classification. However it identifies nearest neighbor instance pairs

Main part is in “Materials and Methods” which contains possible contribution, but anyway it is unclear.

Thank you for pointing this out. We have added sufficient section in material and method to signify the use of the parameters such as  recall, precision, accuracy and f score.

References are not in correct format.

Sure Sir the references have been revisited. All of the references have been updated as per Sensors.

Round 2

Reviewer 1 Report

Accept

Author Response

Thank you so much for accepting our research work.

Reviewer 2 Report

I would like to express my appreciation for authors' thoughtful response to my previous feedback on the manuscript. Your efforts in addressing the concerns raised have undoubtedly improved the overall quality of the paper.

However, upon reviewing the revised version, I noticed that the quality of some images in the manuscript remains suboptimal and hinders their effective interpretation. To ensure the visual clarity and accuracy of the data presented, I kindly request you to make necessary improvements to the images. 

Author Response

Thank you so much. We have updated all the suboptimal images for better visualization. We are also grateful for your feedback which helped us to present our study more efficiently. 

Reviewer 3 Report

-

Many thanks. I checked again the manuscript. The authors fully responded to my comments. The manuscript is in excellent shape, great contribution to the body of science. Please change my recommendation to "accept".

Author Response

Thank you so much. We are also grateful for your valuable feedback.